# Field deployment of a mobile suitcase laboratory for Buruli ulcer diagnosis in Ghana

Hubert Senanu Ahor[1,2], Venus Nana Boakyewaa Frimpong[1], Bernadette Agbavor[1], Kabiru Mohammed Abass[3], George Amofa[4], Elizabeth Ofori[5], Ahmed Abd El Wahed[6], Yaw Ampem Amoako[1,7], Richard Odame Phillips[1,7], Michael Frimpong[1,2]*

1 Kumasi Centre for Collaborative Research, Kwame Nkrumah University of Science and Technology, Kumasi, Ghana, 2 Department of Molecular Medicine, School of Medical Sciences, Kwame Nkrumah University of Science and Technology, Kumasi, Ghana, 3 Agogo Presbyterian Hospital, Agogo, Ghana, 4 Dunkwa Government Hospital, Dunkwa, Ghana, 5 Tepa Government Hospital, Tepa, Ghana, 6 Institute of Animal Hygiene and Veterinary Public Health, University of Leipzig, Leipzig, Germany, 7 Department of Medicine, School of Medical Sciences, Kwame Nkrumah University of Science and Technology, Kumasi, Ghana

* mfrimpong28@gmail.com/frimpong@kccr.de

## Abstract

Molecular diagnostics are the gold standard laboratory confirmation test for Buruli ulcer (BU), a severe necrotising skin disease caused by *Mycobacterium ulcerans (M. ulcerans)*. However, current molecular tests are often performed outside endemic areas, which results in delayed diagnosis and increased patient management costs. To overcome these challenges and facilitate rapid diagnosis of clinically suspected BU lesions in affected communities, we developed a portable laboratory platform contained in two Pelican cases (each measuring 56 cm × 45.5 cm × 26.5 cm). We evaluated the feasibility of performing our earlier developed *M. ulcerans* Recombinase Polymerase Amplification (Mu-RPA) assay, along with a rapid DNA extraction method, using this mobile suitcase laboratory at BU clinics (BU-RPA mobile laboratory) in three endemic districts of Ghana. In the field, the entire process from sample collection to DNA extraction and amplification was completed in under one hour with this mobile setup. Among 39 PCR-confirmed BU cases, 32 (82%; 95% confidence interval [CI]: 67–91) were accurately identified by the BU-RPA mobile laboratory platform. All non-Buruli ulcer cases tested negative, resulting in a clinical specificity of 100% (95% CI: 90–100). Diagnostic performance varied by sample type: swabs demonstrated a sensitivity of 91%, whereas fine-needle aspirates (FNA) had a sensitivity of 69%. This mobile laboratory platform provides an effective workspace for the rapid, on-site diagnosis of BU, enabling timely results for healthcare providers at treatment centres. This mobile suitcase laboratory, together with its isothermal assays, presents a promising alternative to PCR for the swift diagnosis of suspected BU cases as well as other neglected tropical diseases in resource-limited settings.

**Data availability statement:** All data are presented within the submitted manuscript and supporting information.

**Funding:** This work was supported by the European and Developing Countries Clinical Trials Partnership (EDCTP) programme (Grant No. 98684 BU RPA TMA 2015 CDF-979 to M.F). The views and opinions expressed by the authors in this document do not necessarily represent those of EDCTP (http://www.edctp.org). The funders had no role in study design, data collection and analysis, decision to publish, or preparation of the manuscript.

**Competing interests:** The authors have declared that no competing interests exist.

## Author summary

Buruli ulcer, a destructive skin disease caused by *Mycobacterium ulcerans*, is primarily diagnosed through molecular diagnostics. However, these tests are typically conducted in centralized laboratories located outside endemic areas, resulting in delays in diagnosis and treatment. To overcome this challenge, we developed a mobile suitcase laboratory comprising two portable Pelican cases designed for rapid, field-based testing. This setup integrates a field-friendly DNA extraction method with our earlier developed Recombinase Polymerase Amplification assay, which we evaluated in BU clinics across three endemic districts in Ghana. The entire diagnostic process from sample collection to result was completed on-site in under one hour. The mobile laboratory platform accurately identified 82% of PCR-confirmed BU cases and demonstrated 100% specificity for non-BU samples. This portable laboratory offers a reliable and rapid diagnostic tool that could significantly enhance timely BU case management and has the potential to diagnose other neglected tropical diseases in resource-limited settings.

## 1. Introduction

Buruli ulcer (BU), a necrotic skin lesion, caused by *Mycobacterium ulcerans*, primarily affects children under 15, with the majority of cases reported in Africa. The disease begins as nodules, plaques, or edema and progresses to painless ulcers with undermined edges. If left untreated, BU can lead to severe tissue damage, scarring, contractures, and even limb amputation [1,2]. Antimicrobial therapy, involving daily oral administration of rifampicin and clarithromycin, is effective in treating all forms of BU, underscoring the importance of early diagnosis and prompt therapy [1,3].

The World Health Organization (WHO) recommends that at least 70% of suspected BU cases be confirmed through laboratory testing to enhance case management and ensure targeted treatment [4,5]. Polymerase Chain Reaction (PCR) assays targeting the IS2404 sequence of *M. ulcerans* are considered the diagnostic gold standard. However, these assays are often only available at remote reference laboratories, resulting in delays and logistical issues [1]. Furthermore, endemic areas often lack the essential resources and infrastructure needed for molecular diagnosis. These challenges highlight the urgent need for rapid, sensitive, and specific point-of-care diagnostic tools, which align with the WHO's research priorities for Skin neglected tropical diseases [1,4,5]

In response, we developed a rapid molecular test called *M. ulcerans* RPA (Mu-RPA) to detect *M. ulcerans* in clinical samples using an isothermal recombinase polymerase amplification (RPA) assay [6]. This assay amplifies the IS2404 sequence of *M. ulcerans* DNA at 42°C within 15 minutes, with performance varying according to template concentration. It demonstrated a limit of detection of 45 copies of the IS2404 sequence of *M. ulcerans* (corresponding to approximately <1 bacterial

genome equivalent), along with a high diagnostic sensitivity of 88% and a specificity of 100% using PCR as the gold standard [6]. This assay was further refined by optimizing a field-friendly DNA extraction method for the diagnosis of BU in endemic communities [7]. Compared with other molecular assays, this RPA assay offers faster amplification time, compatibility with crude DNA samples, and lower equipment complexity than PCR. In 2018, this assay was proposed as a potential POC diagnostic tool during a consultative meeting organized by the Foundation for Innovative New Diagnostics (FIND) and the World Health Organization/Neglected Tropical Diseases (WHO/NTD) in Geneva [8]. However, this diagnostic assay is yet to be implemented directly at clinics in BU endemic communities due to laboratory infrastructure challenges.

To address the limitations of laboratory infrastructure in communities affected by BU, this study developed a portable suitcase laboratory that facilitates on-site genomic DNA extraction and detection of *M. ulcerans* in suspected BU cases using our MU-RPA assay. This mobile diagnostic platform, known as the BU-RPA mobile laboratory, was tested in three BU clinics in Ghana, to assess the feasibility of operating this platform as well as the diagnostic performance of the Mu-RPA assay in a mobile suitcase in the field, comparing it to the gold standard IS2404 qPCR test performed at the Kumasi Center for Collaborative Research. This center is part of the World Health Organization (WHO) recognized network of laboratories for confirming Buruli ulcer in Africa, the BULABNET [9].

## 2. Materials and methods

**Ethics Statement**: This study received ethical clearance from the Committee on Human Research, Publication and Ethics (CHRPE/AP/122/17) at the School of Medical Sciences, Kwame Nkrumah University of Science and Technology. Written informed consent was obtained from all participants before sample collection. For illiterate participants, study information was read/explained in their local language and consent was documented by thumbprint in the presence of an impartial witness. For children, written consent was obtained from parents or legal guardians, and age-appropriate assent was obtained from the children when applicable. Participation was voluntary, with the right to withdraw at any time. All samples were processed anonymously to ensure patient confidentiality. The individual in this photograph has provided written informed consent, as outlined in the PLOS consent form, to publish this image.

This study was a prospective, field-based diagnostic evaluation that involved the development of a mobile suitcase laboratory and the assessment of a molecular diagnostic assay for patients with clinically suspected BU in treatment clinics.

### 2.1. Development of the mobile suitcase laboratory

The mobile suitcase laboratory was constructed using two Pelican cases (ZARGES GmbH, Weilheim, Germany), each measuring 56 cm × 45.5 cm × 26.5 cm [10,11]. To ensure the safety of the contents during transport, the interiors of these cases were lined with foam. A PVC carpet was attached to the foam, and custom-cut, sealed indents were created to make the cases watertight (S1 Fig). These indents were specifically designed to securely hold all essential materials and equipment for sample processing and amplification, as outlined in Table 1. To minimize the risk of cross-contamination, one suitcase was designated for DNA extraction using a modified Mu-DNA GenoLyse protocol [7], while the other was reserved exclusively for amplification and detection using our already developed Mu-RPA assay [6] (Fig 1A). The lids of the cases were also equipped with compartments for gloves, automatic micropipettes, markers, extension boards, and storage boxes containing buffers and RPA kits. The primary power supply for the laboratory is a GOALZERO Yeti 150 power pack (GOALZERO, South Bluffdale, UT, USA), as shown in Fig 1A. This complete setup including the mobile suitcases, the DNA extraction assay and MU-RPA amplification assay is termed as the BU-RPA mobile laboratory platform.

### 2.2. Field evaluation of the BU-RPA mobile laboratory platform

Molecular diagnosis of clinically suspected BU patients was conducted using the BU-RPA mobile laboratory platform (Fig 1A) at various treatment clinics in three BU-endemic districts in Ghana [12] from 2018 to 2020. These clinics included Agogo Presbyterian Hospital (Asante Akim North District), Tepa Government Hospital (Ahafo Ano North District), and

**Table 1. List of equipment, materials/consumable and reagents in the mobile suitcase laboratory.**

| Equipment |
| --- |
| MyBlock Mini Dry Bath* (Benchmark) |
| Mini Vortex mixer |
| Ika mini G centrifuge |
| WESTMARK Timer |
| Eppendorf pipettes (10 µl, 100 µl, 200 µl and 1ml) |
| Axxin T8 isothermal fluorometer # |
| Mediware disposable scalpels |
| **Material/Consumables** |
| Gloves |
| Pipette tips (10 µl, 100 µl, 200 µl and 1ml) |
| Waste bin |
| Eppendorf tubes (200µl, 2mL) |
| Tissue/paper towel |
| Tube racks |
| Markers |
| Disinfectant swab |
| DNase-free Water |
| Twist Amp Exo kit# |
| GenoLyse extraction kit* |
| Primers and probes# |

# and * items in amplification and extraction suitcase respectively.

Dunkwa Government Hospital (Upper Denkyira East District) (Fig 2). Geographic maps were generated using QGIS version 3.34 LTR (Prizren) with base map data from OpenStreetMap (OpenStreetMap contributors), licensed under the Open Database License (ODbL). These facilities were selected for their accessibility to individuals in rural communities affected by BU. Each site features treatment clinics staffed by trained field workers, nurses, and doctors. Treatment sessions are held weekly on Wednesdays at Agogo, and biweekly on Tuesdays at Dunkwa and Thursdays at Tepa. On clinic days, the mobile suitcase laboratories, along with the power pack and necessary reagents (RPA kits, primers and probes), were transported to these facilities to facilitate on-site testing (Fig 1B). The mobile suitcases were set up at the various BU clinics as shown in Fig 1C on clinic days. A total of seventy-three (73) participants were recruited from all three BU clinics.

**2.2.1. Workflow in the BU-RPA Mobile Laboratory.** The workflow of the BU-RPA mobile Laboratory platform in the field is described below (Fig 3). In the field, the suitcases were positioned with the backs of the lids facing each other to further minimize the risk of cross-contamination (Fig 1C).

**Sample collection:** In the field, swabs and fine-needle aspirates (FNAs) were collected from ulcers and non-ulcerative lesions, respectively, following standard specimen collection procedures at the BU clinics [13]. For each patient (two samples were taken from the same lesion), one sample was taken for Mu-RPA (field diagnosis) and one sample for laboratory confirmation via PCR at the KCCR [14,15], described briefly in the subsequent sections. For DNA extraction and amplification in the field, swab specimens were eluted with 200 µL of GenoLyse lysis buffer, while FNA samples were eluted with 100 µL. These eluted samples were immediately sent to the BU-RPA mobile suitcase platform location. The time of sample collection was noted.

**Extraction suitcase:** The extraction of *M. ulcerans* DNA from all clinical samples collected in the field was carried out in the extraction suitcase using a modified Mu DNA GenoLyse protocol as previously described [7]. In brief, the eluted

PLOS Neglected Tropical Diseases

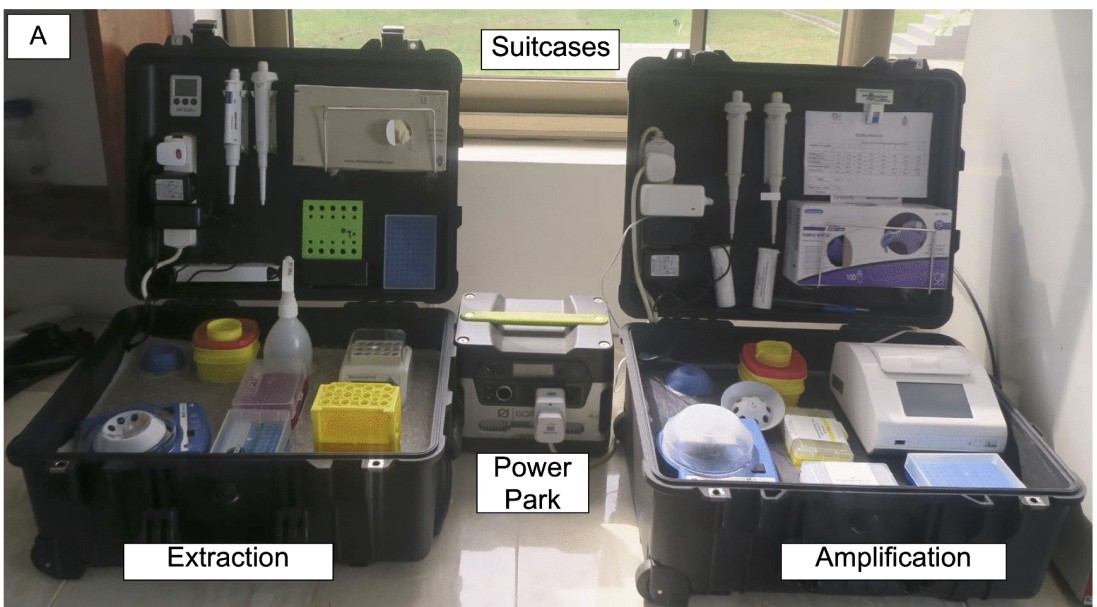

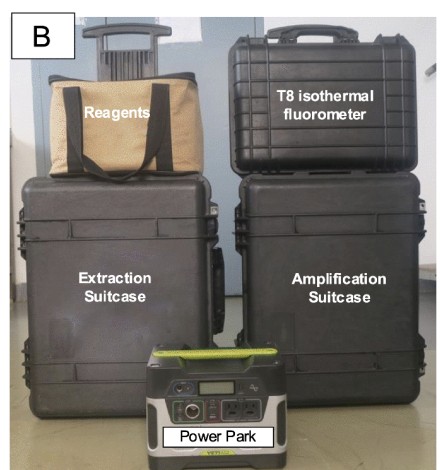
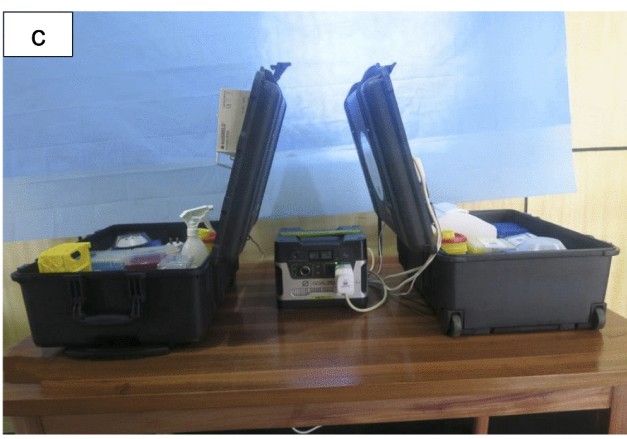

**Fig 1. BU-RPA mobile suitcase laboratory development and deployment in the field. (A)** The fully assembled mobile suitcase laboratory comprises two Pelican cases (ZARGES GmbH, Weilheim, Germany), each measuring 56 cm × 45.5 cm × 26.5 cm and equipped with a power pack. The left suitcase is designed for DNA extraction, while the right suitcase is dedicated to DNA amplification. **(B)** Field deployment of the BU-RPA mobile laboratory platform. The suitcase laboratories, along with the power pack and essential reagents including RPA kits, primers, and probes are routinely transported to BU clinics. **(C)** Field setup of the mobile suitcase laboratory at the BU clinics.

samples were vortexed for 5 seconds, briefly centrifuged, and then incubated at 95°C for 10 minutes. Following incubation, 100 µL of GenoLyse neutralization buffer was added, and the mixture was centrifuged for 5 minutes. A 5 µL aliquot of the resulting supernatant (i.e., nucleic acid extract) was then used for DNA amplification using the previously validated MU-RPA assay [6]. An extraction control that includes only the reagents from the extraction kit, without an actual sample, was included during DNA extraction.

**Amplification suitcase:** Amplification of *M. ulcerans* DNA in the extracted samples was performed using our already developed Mu-RPA assay [6] in the amplification suitcase. In brief, after DNA extraction, 5 µL of extracted DNA template

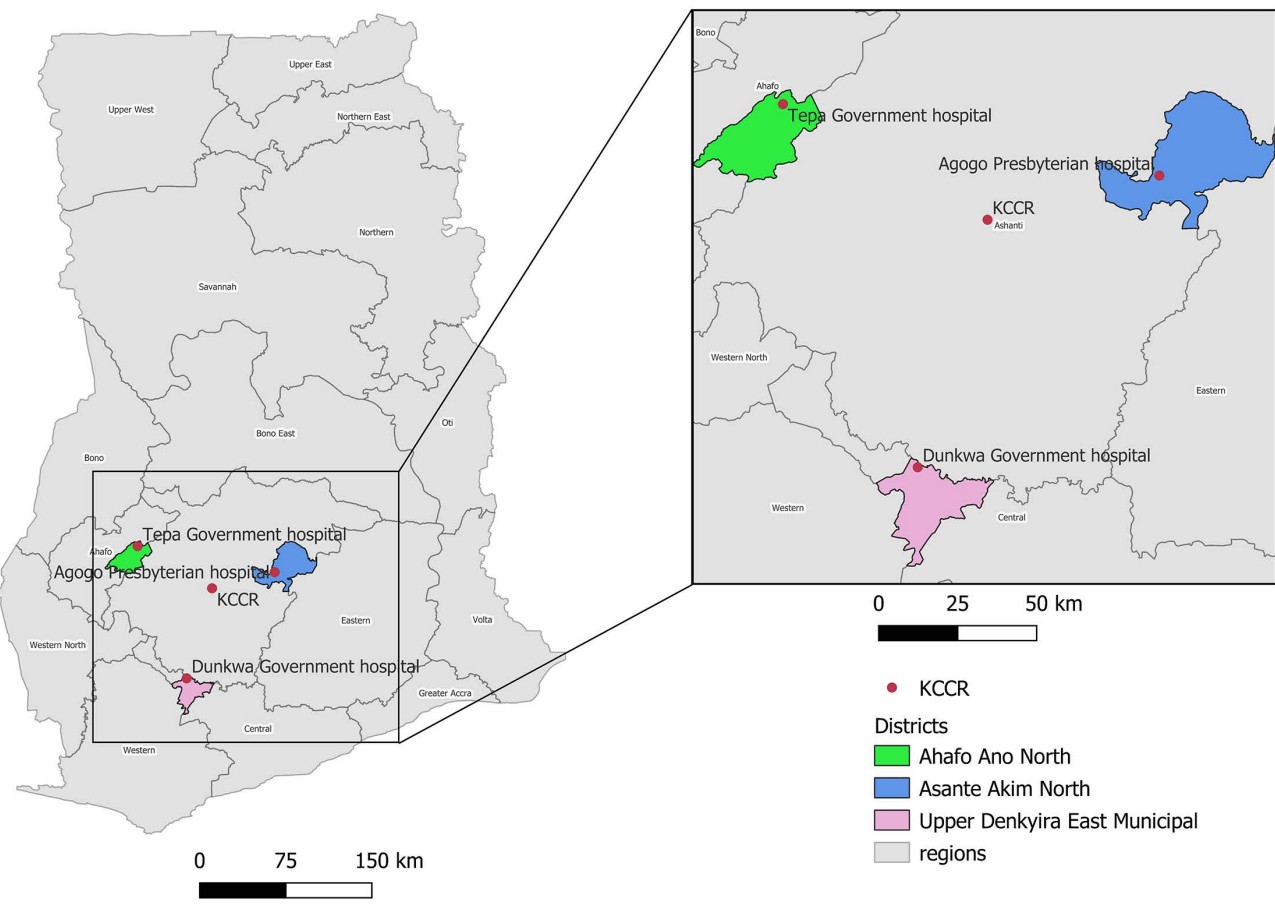

**Fig 2. Map of Ghana showing the study sites.** The map was generated using QGIS version 3.34 LTR (Prizren). Base map data OpenStreetMap contributors, licensed under the Open Database License (ODbL) (https://www.openstreetmap.org).

was added to the reconstituted lyophilized RPA pellet (TwistDx Exo kit, Cambridge, MA, USA). Reconstitution was done with 2.1 µL of 10 µM forward primer, 2.1 µL of 10 µM reverse primer, 0.6 µL of 10 µM probe, 29.5 µL of rehydration buffer, 8.2 µL of DNase-free water, and 2.5 µL of 280 mM magnesium acetate (MgAc). The RPA reaction was incubated at 42°C for up to 15 minutes, and fluorescence signals were detected using an Axxin T8-ISO fluorometer (Axxin Pty Ltd., Victoria, Australia). The Axxin T8-ISO fluorometer displays a positive or a negative result based on the previous optimal program [6]. The time of completion of the assay was also noted. In addition to the extraction control, a "no template control" was added to the amplification reaction to detect contamination of reagents or equipment by extraneous nucleic acids ensuring that results are not due to false positives. Additionally, a positive control, which consists of extracted DNA from *M. ulcerans*, was included to confirm that the RPA system (reagents, primers, polymerase, and amplification machine) was functioning correctly.

### 2.3. Laboratory confirmation of BU cases with qPCR

All suspected BU cases were confirmed by IS2404-targeted qPCR as previously described at KCCR [14,15]. Swab (from ulcerative lesions) and fine needle aspirate (FNA; from non-ulcerative lesions) samples were collected into cell lysis

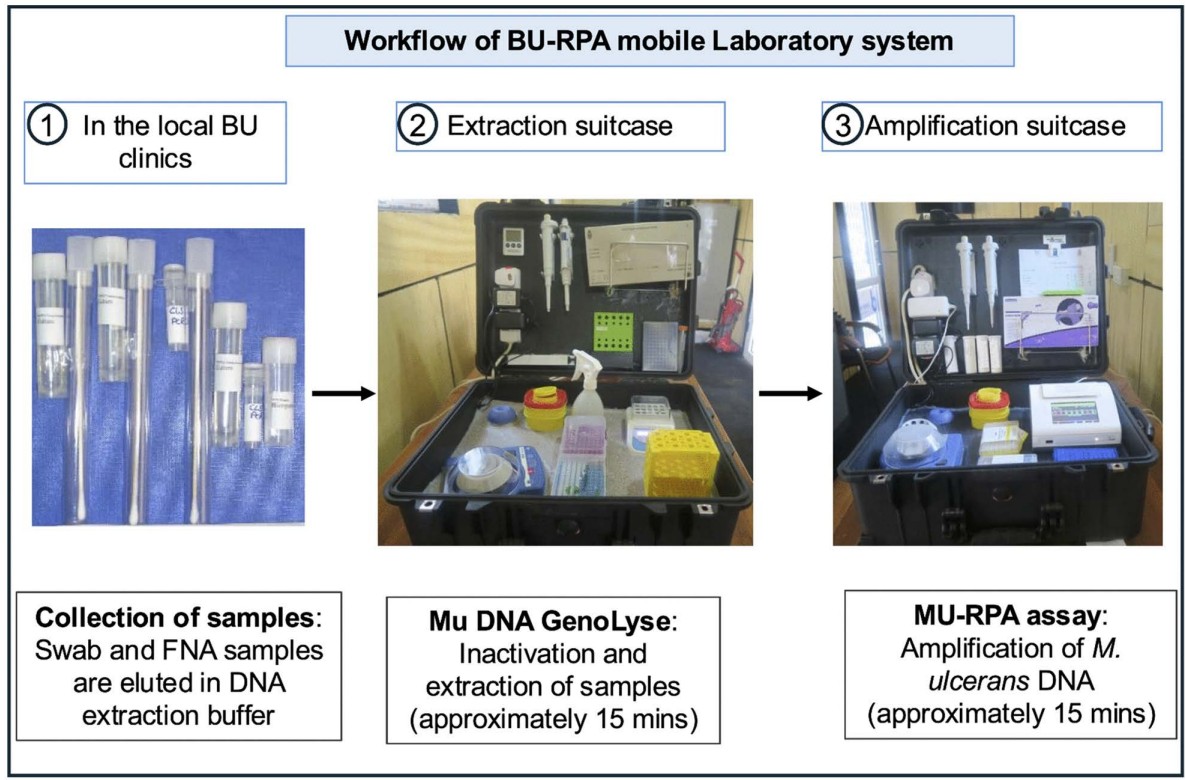

**Fig 3. Workflow of the BU-RPA mobile laboratory platform.** (1) Clinical screening and collect samples of swabs from ulcerative lesions and fine needle aspirates (FNA) from non-ulcerative lesions. Elute samples in GenoLyse lysis buffer and transfer them immediately to the mobile laboratory. (2) Perform rapid DNA extraction in the extraction suitcase using a modified GenoLyse protocol that includes an extraction control. (3) Set up the RPA reaction and conduct isothermal amplification targeting IS2404 in the amplification suitcase, incorporating positive and non-template controls. Use the Axxin T8-ISO fluorometer for real-time fluorescence detection and interpret the results on-site. Compare BU-RPA results with laboratory-based IS2404 qPCR for performance evaluation.

solution (700 µL for swabs; 300 µL for FNA). DNA was extracted using the Gentra Puregene DNA extraction kit (Qiagen, Germany) as described below.

**DNA extraction from clinical samples**: Samples were heat-inactivated at 95°C for 15 minutes, followed by enzymatic digestion with Proteinase K (20 mg/mL) at 55°C for 4 hours or overnight. After inactivation of Proteinase K, samples were treated with lysozyme (10 mg/mL) at 37°C for 1 hour. DNA was then purified using protein precipitation, isopropanol precipitation, and ethanol washing steps. The DNA pellet was air-dried and rehydrated in 200 µL (swabs) or 50 µL (FNA) of hydration solution.

**IS2404 qPCR**: Real-time PCR targeting the IS2404 sequence was performed using the Hot FIREPol Probe qPCR Mix Plus (SolisBioDyne, Estonia) in a 20 µL reaction volume. The reaction contained 1 µl of 5 µM IS2404 TP2 (*5'-FAM-CCGTCCAACGCGATCGGCA-BBQ'-3*), 1 µl each of 10 µM IS2404 TF (*5' AAAGCACCACGCAGCATCT-3'*), and IS2404 TR (*5'-AGCGACCCCAGTGGATTG-3'*) (TibMolBiol), 4 µl of 5 U/µl FIREPol Probe qPCR Mix Plus (SolisBioDyne, Estonia), 2 µl of 10x Exogenous internal positive control (Exo IPC) Mix, 0.4 µl Exo IPC DNA (Invitrogen, UK) and 8.6 µl Diethyl pyrocarbonate (DEPC) treated water (SolisBioDyne, Estonia) as well as 2 µl of the DNA template. Amplification was carried out on a Rotor-Gene Q instrument (Qiagen, Germany) under the following conditions: initial denaturation at 95°C for 15 minutes, followed by 40 cycles of 95°C for 15 seconds and 60°C for 60 seconds. Positive, no-template, and extraction controls were included in each run.

## 2.4. Statistical analysis

Patient data were collected using a standardized BU report form and entered twice into a Microsoft Excel database to ensure accuracy. Additional raw data were also recorded in Microsoft Excel before analysis (S1 File). The data were analyzed with GraphPad Prism v.6 (GraphPad Software, San Diego, CA, USA). General descriptive statistics, including frequencies, percentages, medians, and interquartile ranges, were calculated. Mann-Whitney and Kruskal-Wallis tests were employed to compare the turnaround times in diagnosing types of samples (FNA and swab) and types of lesion (Ulcer, plaque, nodule, edema), respectively. Turnaround time was measured from the time of sample collection to the completion of amplification/result readout. A contingency table was used to calculate the sensitivity, specificity, and predictive values of the Mu-RPA assay, using IS2404 qPCR as the gold standard. To assess the concordance beyond chance between the mobile RPA platform and PCR assays, we utilized Cohen's kappa (κ) statistic, accompanied by 95% confidence intervals and p-values. The κ coefficient was calculated using the epi. kappa function from the epiR package in R (version 4.5.2). Additionally, McNemar's test (R version 4.5.2) was employed to compare the paired proportions (positivity rates) of the RPA and PCR assays. A p-value of less than 0.05 was considered statistically significant.

The sample size was estimated using a single-proportion formula with a 95% confidence level [16]. Assuming an 85% sensitivity and a 95% specificity, we determined that a minimum of 49 PCR-positive cases and 18 PCR-negative cases were required for ±10% precision. Ultimately, 73 participants were enrolled, yielding strong specificity and clinically significant sensitivity estimates, along with reported confidence intervals.

## 3. Results

### 3.1. BU-RPA mobile laboratory platform

A mobile diagnostic system was created using two Pelican cases (56 cm × 45.5 cm × 26.5 cm) to enable on-site DNA extraction and Recombinase polymerase amplification (Fig 1A). Powered by a Goal Zero YETI 400 portable power pack (GOALZERO, South Bluffdale, USA), the system operates independently in field settings. This configuration enabled the successful implementation of direct DNA extraction and amplification from clinically suspected BU lesions at BU treatment clinics.

### 3.2. Demographic and clinical characteristics of study participants

The study included a total of 73 clinically suspected cases of BU. Female participants made up the majority at 61.6%, while males accounted for 38.4%. The median age of participants was 18 years, with a range of 1–77 years. Most lesions were classified as ulcers (60.3%), followed by plaques (23.3%) and nodules (13.7%). In terms of lesion category classification, nearly half of the lesions were categorized as category I (49.3%), while categories II and III accounted for 28.8% and 21.9%, respectively (Table 2).

### 3.3. Diagnostic performance of BU-RPA mobile suitcase laboratory under field conditions

The BU-RPA mobile suitcase laboratory comprising of suitcases, Mu-RPA assay, along with the Mu GenoLyse DNA extraction protocol [6,7], was evaluated in at clinics in BU-endemic districts (Fig 1). The average diagnostic turnaround time from specimen collection to amplification completion was approximately 45 minutes (Fig 4). The PCR assay performed in a reference laboratory, which took at least two days. The turnaround time of BU diagnosis in the field was not affected by the type of sample (Fig 4A) or BU lesions (Fig 4B).

Of 73 patients suspected of having BU (Table 3), 39 were PCR-positive. The BU-RPA mobile laboratory platform accurately detected 32 of the 39 confirmed cases, yielding a sensitivity of 82% (95% Confidence Interval (CI): 67–91%). Additionally, all 34 patients without BU were correctly identified as negative, resulting in a specificity of 100% (95% CI: 90–100%). When analyzed by specimen type, sensitivity was 69% (95% CI: 44–86) for FNA samples and 91% (95% CI: 73–98) for swab samples, with specificity remaining at 100% for both sample types (Table 3).

**Table 2. Demographic and clinical characteristics of suspected BU cases.**

| Characteristics | | No. (%) of total lesions (n=73) |
|---|---|---|
| **Sex** | Male | 28 (38.4) |
| | Female | 45 (61.6) |
| **Age (years)** | Median (Range) | 18 (1 - 77) |
| **Sample type** | Swab | 44 (60.3) |
| | FNA | 29 (39.7) |
| **Type of lesion** | Ulcer | 44 (60.3) |
| | Nodule | 10 (13.7) |
| | Plaque | 17 (23.3) |
| | Edema | 2 (2.7) |
| **Category of lesion** | I | 36 (49.3) |
| | II | 21 (28.8) |
| | III | 16 (21.9) |

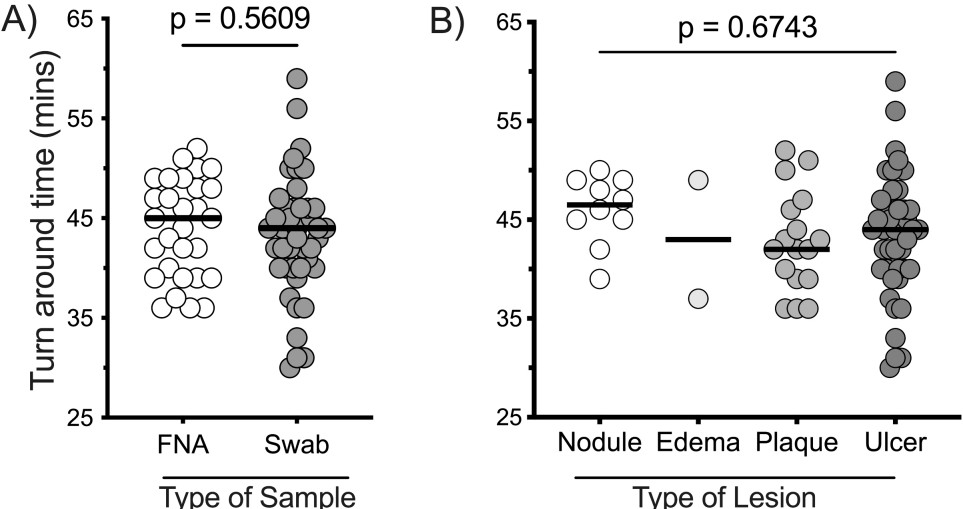

**Fig 4. Turn around time of the BU-RPA mobile laboratory platform.** The turnaround time for diagnosing clinically suspected BU lesions using the BU-RPA mobile laboratory platform, stratified by sample type **(A)** and lesion type **(B)**. Mann-Whitney and Kruskal-Wallis tests were employed to compare the turnaround time in diagnosing types of samples (FNA (n=29) and swab (n=44) and types of lesions (ulcer (n=44), plaque (n=17), nodule (n=10), edema (n=2)) respectively, with a p-value of less than 0.05 considered statistically significant.

Across all 73 samples, the BU-RPA mobile laboratory platform demonstrated a high level of agreement with qPCR, achieving an observed agreement of 90%. This strong concordance was further validated by Cohen's kappa ($\kappa=0.81$, 95% CI: 0.58–1.00, $p<0.001$). Overall positivity rates were higher for qPCR at 53.4% compared to BU-RPA mobile laboratory platform at 43.8%, with this difference being statistically significant (McNemar's test, $p=0.008$). In swab samples, the agreement was even greater, with an observed agreement of 96% and nearly perfect concordance ($\kappa=0.91$, 95% CI: 0.62–1.00, $p<0.001$). The positivity rates were similar for qPCR (52.3%) and BU-RPA mobile laboratory platform (47.7%), and this difference was not statistically significant (McNemar's test, $p=0.48$). For FNA samples, the agreement was lower but still substantial, with an observed agreement of 82.8% and $\kappa=0.66$ (95% CI: 0.32–1.00, $p<0.001$). The positivity rates

PLOS Neglected Tropical Diseases

**Table 3. Clinical sensitivity and specificity of the BU-RPA mobile laboratory platform compared to qPCR (stratified by characteristic of BU lesions and sample type).**

| | | PCR+ve | PCR –ve | Total | Sensitivity % (95% CI) | Specificity % (95% CI) | PPV % (95% CI) | NPV % (95% CI) | Kappa | P-value* |
|---|---|---|---|---|---|---|---|---|---|---|
| **Swab** | BU-RPA+ve | 21 | 0 | 21 | 91 (73-98) | 100 (85-100) | 100 (85-100) | 91 (73-98) | | |
| | BU-RPA -ve | 2 | 21 | 23 | | | | | 0.91 | P<0.0001 |
| | Total | 23 | 21 | 44 | | | | | | |
| **FNA** | BU-RPA+ve | 11 | 0 | 11 | 69 (44-86) | 100 (77-100) | 100 (74-100) | 72 (49-88) | | |
| | BU-RPA -ve | 5 | 13 | 18 | | | | | 0.66 | P<0.0001 |
| | Total | 16 | 13 | 29 | | | | | | |
| | BU-RPA+ve | 32 | 0 | 32 | 82 (67-91) | 100 (90-100) | 100 (89-100) | 83 (69-91) | | |
| **Overall** | BU-RPA -ve | 7 | 34 | 41 | | | | | 0.81 | P<0.0001 |
| | Total | 39 | 34 | 73 | | | | | | |

**\*** A p-value of less than 0.05 is considered statistically significant.

were higher for qPCR at 55.2% compared to RPA at 37.9%, although this difference did not reach statistical significance (McNemar's test, p=0.074). Overall, these findings suggest that while BU-RPA mobile laboratory platform performs well in comparison to qPCR, its sensitivity was reduced in FNA samples.

## 4. Discussion

Developing field-deployable diagnostic tools for neglected tropical diseases (NTDs) such as Buruli ulcer (BU) remains a key research priority [1,17]. To support this objective, our research group established a field-friendly Mu DNA GenoLyse extraction technique and a real-time recombinase polymerase amplification assay for the rapid detection of *M. ulcerans*, the causative agent of BU [6,7]. To facilitate the effective implementation of these assays in BU endemic regions, we developed the BU-RPA mobile laboratory platform in this study to enable DNA extraction and Amplification at BU clinics in endemic communities. This platform was transported on clinic days and evaluated at three BU clinics in Agogo Presbyterian Hospital, Tepa District Hospital, and Dunkwa Government Hospital for on-site diagnosis of clinically suspected BU cases.

The time from sample collection to nucleic acid amplification ranged from 30 to 60 minutes, comparable to other studies assessing similar platforms for viral disease diagnosis in low-resource settings [11,18,19]. The BU-RPA mobile laboratory platform using our developed Mu-RPA assay achieved a turnaround time of less than 60 minutes, notably faster than BU-LAMP assays (which take over 1 hour) [20–22], fluorescent thin-layer chromatography (fTLC, approximately over 1 hour) [23,24], and the gold standard PCR method (which takes at least 2 days) for BU diagnosis [5,25]. While the Mu-RPA amplification required only 15 minutes, the total turnaround time of approximately 45 minutes was largely due to the DNA extraction step; further streamlining of extraction could reduce the overall time to under 30 minutes." The average turnaround time of about 45 minutes was mainly because one operator was handling both suitcases. Using two trained personnel could expedite processing and reduce the risk of contamination. In this study, the BU-RPA mobile laboratory platform was operated by personnel trained in advanced molecular diagnostics. For successful implementation in resource-limited settings, structured and targeted training will be crucial for non-specialist. This training must empower non-specialist users, including healthcare workers and disease control officers to competently carry out all essential steps, from sample preparation and DNA extraction to amplification and result interpretation. With the right training and standard operating procedures in place, the platform can be effectively utilized at decentralized levels of the healthcare system.

A significant challenge for molecular diagnostics in low-resource, disease-endemic regions is maintaining a reliable power supply [26]. To address this issue, we utilized the Goal Zero YETI portable power pack to operate the mobile laboratory [10,11]. This device provided uninterrupted electricity, could be recharged using solar panels or mains power, and offered power for 10–12 hours [10]. This demonstrates the feasibility of conducting molecular diagnostics using isothermal DNA amplification methods, such as RPA, in endemic areas with unstable electricity supplies. Other studies have also used car batteries as alternative power sources for mobile laboratories [18,27].

Under field conditions, the BU-RPA mobile laboratory platform demonstrated a clinical sensitivity of 82% and a specificity of 100%. The reduced sensitivity was primarily due to lower positivity rates in FNA samples, while swab samples showed higher sensitivity. Swabs collected from ulcerative lesions generally contain higher bacterial loads, which likely accounts for their better detection performance. In contrast, FNA samples often contain red blood cells and tissue-derived proteins, which can compromise DNA quality and yield, potentially lowering IS2404 copy numbers below the assay's detection threshold. The lower agreement and sensitivity in FNA samples may also result from the inherently lower bacterial burden in non-ulcerative lesions and variability in sampling when different aspirates from the same lesion are used for PCR and RPA testing. Additionally, the crude Mu-GenoLyse extraction method employed under field conditions may introduce inhibitors, such as host DNA and proteins, leading to false-negative RPA results. To improve performance, it is crucial to optimize sample collection and processing. This includes standardizing sampling from the same lesion site, refining DNA extraction methods to minimize inhibitors, and using diluted DNA templates when appropriate. Repeat testing or re-sampling may also help reduce false-negative results, particularly for FNA specimens. Together, these measures could enhance the sensitivity and reliability of the RPA assay in non-ulcerative lesions.

The BU-RPA mobile laboratory platform demonstrates superior diagnostic performance compared to other field-deployable assays, such as BU-LAMP and fluorescent thin-layer chromatography (fTLC), which showed clinical sensitivity ranging from 25% to 80% and specificity from 35% to 75% in Ghanaian district hospitals with both swab and FNA samples [23,24]. While the diagnostic performance of BU-RPA mobile laboratory in field conditions is lower than the performance of MU-RPA in laboratory settings (88% sensitivity and 100% specificity) [6,7] and our previously developed Biomeme Franklin Mobile qPCR, which achieved 97% to 100% sensitivity and 94% to 100% specificity in laboratory conditions [28], the BU-RPA mobile laboratory platform still remains robust in the field. Notably, its diagnostic accuracy surpasses the WHO minimum performance threshold of ≥70% laboratory confirmation for BU, reinforcing its value as a reliable field-deployable diagnostic tool [21]. Similar mobile RPA-based platforms have been successfully utilized for other infectious diseases, including Ebola in Guinea [27], dengue virus in Senegal [18] and leishmaniasis in Sri Lanka [19] reported sensitivities ranging between 65–98%. The recent increase in the development of RPA assays for various diseases [29] promotes the use of mobile laboratory suitcases to extend molecular diagnostics into low-resource settings. This approach has the potential to enable early case detection and timely treatment, ultimately reducing the burden of neglected tropical diseases.

The estimated total cost of the mobile suitcase laboratory, including the power pack, is approximately $10,000. The cost per test for the RPA assay is around $10, which is lower than the estimated $20–25 per test for conventional or portable PCR platforms, such as the Biomeme Franklin mobile qPCR. The higher expenses associated with portable PCR platforms are primarily due to proprietary DNA extraction kits. This estimate covers the costs of RPA reagents, DNA extraction kits, and consumables but excludes capital equipment and personnel costs, reflecting the direct operational cost per test.

This cost advantage, along with the simplified workflow of RPA, makes it particularly suitable for resource-limited, high-endemicity settings. Compared to portable qPCR systems, the RPA-based approach offers several operational benefits (S1 Table), including rapid amplification (approximately 15 minutes), the elimination of thermal cycling, and compatibility with crude extracted DNA extract and alternative detection formats like lateral flow readouts. These features allow for the use of simpler instrumentation and support deployment in decentralized settings with limited infrastructure [11]. While a per-test cost of $10 may still present challenges for large-scale implementation, further reductions could be achieved

through bulk reagent production, local manufacturing, simplified DNA extraction methods, and increased market competition. Additionally, incorporating an internal positive control and a quantification module could enhance the assay's utility for treatment monitoring.

Deploying the BU-RPA mobile laboratory platform at district-level hospitals could eliminate the need to transport samples to distant reference laboratories, thereby reducing diagnostic delays and minimizing loss to follow-up. Importantly, this platform has the potential to align with WHO priorities by serving as a shared diagnostic tool for multiple skin NTDs, including Buruli ulcer. For instance, an RPA assay for yaws has already been developed [30] and could be integrated into the same platform. The versatility of this approach has also been demonstrated in other contexts, where similar mobile laboratory systems were repurposed alongside portable PCR platforms for diagnosing suspected COVID-19 cases during the pandemic [31].

It is also important to note that the study included fewer PCR-positive cases (n = 39) than initially estimated in the sample size calculation, which may affect the precision of diagnostic performance estimates. Overall, these findings highlight the potential of the RPA platform as a practical, cost-effective, and field-adapted alternative to PCR, while emphasizing the need for further optimization and larger-scale evaluation.

## 5. Conclusion

This study highlights the effective use of the BU-RPA mobile laboratory platform for the rapid molecular confirmation of Buruli ulcer in endemic areas. Building on prior analytical validation, our findings offer real-world evidence of this RPA-based approach's feasibility, robustness, and diagnostic performance in field conditions. The platform enhances workflow efficiency, reduces turnaround time, and lowers per-test costs compared to conventional PCR methods, making it a promising option for decentralized, resource-limited healthcare settings. However, despite its strong overall performance, further optimization especially for FNA samples is needed to enhance sensitivity before large-scale implementation. The BU-RPA mobile laboratory platform thus serves as a complementary, field-adapted diagnostic solution that aligns with WHO priorities for improving access to timely Buruli ulcer diagnoses. Future efforts should concentrate on larger-scale implementation, evaluation in routine healthcare settings, assessment of user training needs, comparative analyses with portable qPCR systems in the field, and cost optimization to facilitate sustainable integration into national control programs.

## Supporting information

**S1 Fig. Assembly of the Mobile Suitcase Laboratory.**
(TIF)

**S1 File. Raw data of the study.**
(XLSX)

**S1 Table. Comparison of molecular diagnostic platforms for Buruli ulcer detection.**
(DOCX)

## Acknowledgments

We would like to express our gratitude to all study participants for their involvement. We also appreciate the collaboration of the nurses and physicians at the Buruli ulcer clinics in Tepa Government Hospital, Dunkwa Government Hospital, and Agogo Presbyterian Hospital. Special thanks are extended to Mr Jonathan Kofi Adjei, Mrs Abigail Agbanyo, Mr Michael Ntiamoah Oppong, Mr Wilfred Aniagyei and Mr Samuel Opoku for their exceptional technical support.

## Author contributions

**Conceptualization:** Richard Odame Phillips, Michael Frimpong.

**Data curation:** Hubert Senanu Ahor, Venus Nana Boakyewaa Frimpong, Bernaddette Agbavor, Yaw Ampem Amoako, Michael Frimpong.

**Formal analysis:** Hubert Senanu Ahor.

**Funding acquisition:** Richard Odame Phillips, Michael Frimpong.

**Investigation:** Hubert Senanu Ahor, Venus Nana Boakyewaa Frimpong, Bernaddette Agbavor, Kabiru Mohammed Abass, George Amofa, Elizabeth Ofori.

**Methodology:** Hubert Senanu Ahor, Bernaddette Agbavor, Yaw Ampem Amoako, Michael Frimpong.

**Project administration:** Michael Frimpong.

**Resources:** Bernaddette Agbavor, Ahmed Abd El Wahed, Yaw Ampem Amoako, Richard Odame Phillips.

**Supervision:** Ahmed Abd El Wahed, Richard Odame Phillips.

**Validation:** Ahmed Abd El Wahed, Yaw Ampem Amoako.

**Writing – original draft:** Hubert Senanu Ahor, Michael Frimpong.

**Writing – review & editing:** Hubert Senanu Ahor, Venus Nana Boakyewaa Frimpong, Bernaddette Agbavor, Kabiru Mohammed Abass, George Amofa, Elizabeth Ofori, Ahmed Abd El Wahed, Yaw Ampem Amoako, Richard Odame Phillips, Michael Frimpong.

## Refences

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
