## [Decision Letter · Decision Letter 0]

3 Dec 2025

PNTD-D-25-01865

Deployment of a mobile suitcase laboratory for field diagnosis of Buruli ulcer disease in Ghana

Dear Dr. Frimpong,

Thank you for submitting your manuscript to PLOS Neglected Tropical Diseases. After careful consideration, we feel that it has merit but does not fully meet PLOS Neglected Tropical Diseases's publication criteria as it currently stands. Therefore, we invite you to submit a revised version of the manuscript that addresses the points raised during the review process.

Please submit your revised manuscript within by Jan 31 2026 11:59PM. If you will need more time than this to complete your revisions, please reply to this message or contact the journal office at plosntds@plos.org. Please include the following items when submitting your revised manuscript:

We look forward to receiving your revised manuscript.

Kind regards,

Anil Fastenau, M.D., M.Sc.

Guest Editor

Mathieu Picardeau

Section Editor

Shaden Kamhawi

co-Editor-in-Chief

Paul Brindley

co-Editor-in-Chief

Journal Requirements:

1) We noticed that you used the phrase 'data not shown' in the manuscript. We do not allow these references, as the PLOS data access policy requires that all data be either published with the manuscript or made available in a publicly accessible database. Please amend the supplementary material to include the referenced data or remove the references.

- ® on pages: 5, 8, 10, and 12.

3) 2E includes an image of an identifiable person. Please confirm that you have obtained written confirmation or release forms, signed by the subject(s) (or their guardian), giving permission to be photographed and to have their images published under a Creative Commons license. You can find more details on the journal's informed consent policy and a downloadable Consent Form for Publication in a PLOS Journal here: https://journals.plos.org/plosone/s/human-subjects-research#loc-patient-privacy-and-informed-consent-for-publication.

Please do NOT upload the signed form as part of your submission files, in order to protect the privacy of the patient's identity. Instead, in the Ethics Statement in the Methods section of your manuscript, please confirm that the patient has provided consent by inserting a statement such as the following:

"The individual in this photograph has given written informed consent (as outlined in PLOS consent form) to publish this image."

If you have not received consent from the photograph subject to publish their image, please remove the image or blur or crop the individual in the photo so that their face is not visible.

Potential Copyright Issues:

i) Please confirm (a) that you are the photographer of 1, 2A, 2C, 2D, 2E, and 3, or (b) provide written permission from the photographer to publish the photo(s) under our CC BY 4.0 license.

ii) Figure 2B. Please (a) provide a direct link to the base layer of the map (i.e., the country or region border shape) and ensure this is also included in the figure legend; and (b) provide a link to the terms of use / license information for the base layer image or shapefile. We cannot publish proprietary or copyrighted maps (e.g. Google Maps, Mapquest) and the terms of use for your map base layer must be compatible with our CC BY 4.0 license.

5) We note that your Data Availability Statement is currently as follows: "All data are presented within the submitted manuscript.". Please confirm at this time whether or not your submission contains all raw data required to replicate the results of your study. Authors must share the “minimal data set” for their submission. PLOS defines the minimal data set to consist of the data required to replicate all study findings reported in the article, as well as related metadata and methods (https://journals.plos.org/plosone/s/data-availability#loc-minimal-data-set-definition).

2) If the funders had no role in your study, please state: "The funders had no role in study design, data collection and analysis, decision to publish, or preparation of the manuscript.".

Reviewers' Comments:

Reviewer's Responses to Questions

Key Review Criteria Required for Acceptance?

Methods

-Are the objectives of the study clearly articulated with a clear testable hypothesis stated?

-Is the study design appropriate to address the stated objectives?

-Is the population clearly described and appropriate for the hypothesis being tested?

-Is the sample size sufficient to ensure adequate power to address the hypothesis being tested?

-Were correct statistical analysis used to support conclusions?

-Are there concerns about ethical or regulatory requirements being met?

Reviewer #1: -Are the objectives of the study clearly articulated with a clear testable hypothesis stated?no

-Is the study design appropriate to address the stated objectives?no

-Is the population clearly described and appropriate for the hypothesis being tested?yes

-Is the sample size sufficient to ensure adequate power to address the hypothesis being tested? no sample size described

-Were correct statistical analysis used to support conclusions? yes

-Are there concerns about ethical or regulatory requirements being met?no

Reviewer #2: • Clarify whether the two samples used for both Mu-RPA and PCR were collected at the same site of the lesion or from different sites. This may affect interpretation of sensitivity and specificity.

• Add details on contamination control measures and internal quality controls.

• Specify operator training level and discuss feasibility for non-specialists.

• How did authors manage contentment for children? To the include in the manuscript

• Sample size is too small.

Reviewer #3: -Are the objectives of the study clearly articulated with a clear testable hypothesis stated?

The hypothesis and specific objectives could be stated more clearly in the end of the introduction. What did the authors expect as outcome, which outcome measures were focussed on?

-Is the study design appropriate to address the stated objectives?

Yes.

-Is the population clearly described and appropriate for the hypothesis being tested?

Yes, however, a description of the study setting is missing (prevalence, incidence) and could be included in the introduction. Where the PCR-negative patients diagnosed with other diseases? Could the authors explain why no healthy controls were included?

-Is the sample size sufficient to ensure adequate power to address the hypothesis being tested?

No sample size calculation was included. As no prevalence data is included, it cannot be assessed whether the sample size is sufficient.

-Were correct statistical analysis used to support conclusions?

Major: The statistical analyses are not outlined in the methods section. The authors should add this to the methods section, as well as a sample size calculation to motivate the number of analysed samples.

-Are there concerns about ethical or regulatory requirements being met?

Could the authors expand on how informed consent was obtained from children and illiterate persons, if applicable?

-Please include the positivity cutoff of the PCR and RPA and how it was determined for the RPA assay. What is the limit of detection of this assay?

-Were the samples tested in triplicates? If not, please motivate why. Was a non-template control included, how was controlled for contamination?

Results

-Does the analysis presented match the analysis plan?

-Are the results clearly and completely presented?

-Are the figures (Tables, Images) of sufficient quality for clarity?

Reviewer #1: -Does the analysis presented match the analysis plan? yes but analysis plan should be aligned with MIQE guidelines and should be more elaborate, below I suggest additional experiments

-Are the results clearly and completely presented?yes, but raw data from RPA should be added in the article or in supplemental

-Are the figures (Tables, Images) of sufficient quality for clarity? quality is very low, blurry

Reviewer #2: The main limitation is the sample size.

Reviewer #3: For the comparison of turnaround time between PCR and RPA there is no statistical test performed and no standard deviation in indicated. Did the authors also check whether samples that were stored in the buffer shorter or longer than the average time are more or less positive for M. ulcerans DNA detection? Was the significance of the influence of type of sample or BU lesion tested with a statistical test? Please add in the methods section which test was used and what the cutoff for hypothesis acceptance was.

I suggest adding a comparison of the positivity of PCR and RPA, to understand whether the observed RPA assay sensitivity is significantly different to the gold standard (paired testing).

Figure 1 + 2 could be merged in a general figure and reduced to Figures 1F, 2A (with measurements?), 2C, and 2B.

Figure 3 could be developed further, I recommend depicting the workflow in the different suitcases in more detail. As is, the figure does not have much additional value to the text.

Figure 4: It could be helpful to include the sample number per condition in the figure

Table 3: for FNA, should the sensitivity not be 32/39 = 82 %? The table could benefit from adding a line of total positives and negatives (for swab, FNA, and total), to better understand where the sensitivity calculations come from.

Conclusions

-Are the conclusions supported by the data presented?

-Are the limitations of analysis clearly described?

-Do the authors discuss how these data can be helpful to advance our understanding of the topic under study?

-Is public health relevance addressed?

Reviewer #1: Nice overview of the suitcase, however the picture quality should be optimized

Step-by-step protocol are not included

limitation are not mentioned

Reviewer #2: Major Revision:

The manuscript shows promise but requires substantial improvements before it can be considered for publication. Specifically:

The methodology section needs clearer description and justification of the approach used.

The sample size should be increased or adequately justified to strengthen the validity of the findings.

The discussion should be expanded to include limitations of the study and potential future applications of the findings.

References need a comprehensive review to ensure accuracy, consistency, and proper alignment with in-text citations.

Reviewer #3: Are the conclusions supported by the data presented?

Yes.

Are the limitations of analysis clearly described?

- The authors should describe if healthy controls were included and if not, why that was the case.

- Some of the patients diagnosed by the PCR assay were not detected by the BU-RPA. The authors mention that a higher limit of detection could be the reason. This should be put in the context of previously determined LODs with this assay and also other reasons should be considered, e.g. DNA extraction efficiency that could be influenced by the thick mycobacterial cell wall.

- A further explanation why the FNA sample matrix may be detrimental to the detection efficiency would be helpful. Swabs were only taken from ulcerating lesions, could there be a relationship of the increased sensitivity with higher numbers of bacteria in this type of lesion? How could the sensitivity of FNA samples be increased?

- Please elaborate if the other assays BU-LAMP and fTLC were also done on swabs and FNAs.

- Could the authors include whether the reagents used in the DNA extraction need special storage conditions or are they stable at ambient temperature?

- The authors mention that no internal positive control and quantification was included. Could they expand on why this was left out and how it can be confidentially determined that the negative results in the BU-RPA were not due to extraction or assay failure?

- If samples are not collected in the hospital but more remotely, could the time between sample collection and start of extraction influence the result? Would it be possible to perform the test on inactivated samples as well (e.g. with ethanol, due to biosafety considerations)?

Do the authors discuss how these data can be helpful to advance our understanding of the topic under study?

Yes.

Is public health relevance addressed?

Yes.

Editorial and Data Presentation Modifications?

Reviewer #1: Nice overview of the suitcase, however the picture quality should be optimized

Reviewer #2: (No Response)

Reviewer #3: - Authors seem to use Mu-RPA and BU-RPA interchangeably, please streamline.

- Line 266: there is a word missing after ‘various’

Summary and General Comments

Reviewer #1: There is a field friendly tool available which has been published from which the first author is the last author of the current proposed manuscript (https://pubmed.ncbi.nlm.nih.gov/37228126/) .

This article weakens the claim of originality, as it already presents a field-friendly molecular confirmation method for BU. Although the previously published tool relies on qPCR and the current study focuses on RPA, two different amplification technologies with distinct operational formats,they ultimately serve the same purpose. Therefore, the added value of the new RPA-based test must be clearly articulated in the manuscript, and a direct comparison with the published qPCR method should be included. This is currently missing and should be addressed to demonstrate the specific advantages and justification for the new approach.

The current data and analyses do not yet fully support the claims, as the technical validation of the assay is missing and only the in-field evaluation is presented. I recommend consulting the MIQE guidelines to identify the essential technical experiments that should be added, including determination of the limit of detection (LOD), technical reproducibility, repeatability, and additional control reactions. The inclusion of the EQA BU LabNet panel would further strengthen validation and provide an external benchmark.

Several necessary controls are also lacking: sampling controls, DNA extraction controls (both positive and negative), and RPA assay controls (positive and negative). These should be incorporated to ensure assay reliability and to meet standard reporting expectations.

Given the existence of a the Biomeme Franklin Mobile qPCR for rapid detection of M. ulcerans, a direct comparison is needed. This includes comparing both the detection method and the rapid DNA extraction solution (Genolyse vs Biomeme M1 cartridges) against the rapid extraction protocol proposed here. A side-by-side comparison of DNA extraction methods using the standard IS2404 qPCR as the reference outcome would provide robust evidence for the added value of the new method.

A cost comparison covering both the DNA extraction workflow and the RPA/Biomeme detection approaches should be included to clearly position the practicality and field relevance of the proposed solution.

Also the number of steps can be included as a comparison , as each additional step is a possible chance of contamination , which is something you would like to avoid when you are molecularly confirming cases.

Additinoaaly it would be helpful to also mentioned the swabs that were used. these can give a tremendous difference, cotton swab are preferably not used but floqswab are perfect. they release the capture pathogens way better then cotton swabs.

Reviewer #2: 1. Title: Deployment of a mobile suitcase laboratory for field diagnosis of Buruli ulcer disease in Ghana

Shorten for clarity and impact:

Suggested: “Field deployment of a mobile suitcase laboratory for Buruli ulcer diagnosis in Ghana.”

2. Overall assessment

The manuscript presents an innovative approach to improving Buruli ulcer (BU) diagnosis through a mobile suitcase laboratory using Recombinase Polymerase Amplification (RPA). The study addresses a critical gap in point-of-care molecular diagnostics for neglected tropical diseases (NTDs) and demonstrates feasibility in field conditions. The work is relevant, timely, and potentially impactful for resource-limited settings. However, some areas require clarification, refinement, and additional detail to strengthen the manuscript.

3. Materials and Methods

• Clarify whether the two samples used for both Mu-RPA and PCR were collected at the same site of the lesion or from different sites. This may affect interpretation of sensitivity and specificity.

• Add details on contamination control measures and internal quality controls.

• Specify operator training level and discuss feasibility for non-specialists.

• How did authors manage contentment for children? To the include in the manuscript

• Sample size is too small.

4. Results

If possible, include user feedback from healthcare workers on ease of use and acceptability.

5. Discussion

• Expand on reasons for lower sensitivity in FNA samples and propose solutions.

• Discuss cost implications more thoroughly, feasibility of $10/test in endemic settings.

• Add limitations explicitly: small sample size, single-country evaluation, lack of multicenter validation.

• Include future directions: integration with WHO NTD roadmap, multiplexing potential, digital reporting tools.

6. References

References need major review. The references require a thorough review for accuracy and consistency. In the discussion section, the statement “Developing field-deployable diagnostic tools for neglected tropical diseases (NTDs) such as Buruli ulcer (BU) remains a key research priority for the World Health Organization (WHO) [1, 13]” implies that references 1 and 13 are WHO documents. However, in the reference list, these citations correspond to Yotsu RR et al. and Souza AA et al., respectively, and are not WHO sources. Please ensure that all in-text citations accurately reflect the referenced documents and revise either the text or the references to maintain consistency.

7. Language and Style

• Correct typographical errors (e.g., “Sevety-three” → “Seventy-three”). Line 186

• Use consistent terminology: “BU-RPA mobile laboratory platform” throughout.

8. Additional enhancements

• Consider adding a cost-benefit analysis comparing this platform to PCR and other alternatives.

9. Recommendation

The manuscript shows promise but requires substantial improvements before it can be considered for publication. Specifically:

• The methodology section needs clearer description and justification of the approach used.

• The sample size should be increased or adequately justified to strengthen the validity of the findings.

• The discussion should be expanded to include limitations of the study and potential future applications of the findings.

• References need a comprehensive review to ensure accuracy, consistency, and proper alignment with in-text citations.

Reviewer #3: The manuscript describes the clinical validation of an isothermal assay (Mu-RPA) for the detection of M. ulcerans DNA using the specific IS2404 target. Specifically, they were using a field friendly assay setup that was split over two portable suitcases including both DNA extraction and amplification/read-out. Analysing different clinical samples from suspect BU patients, the authors found a sensitivity of this RPA assay of 83 % compared to the diagnostic gold standard of PCR detection and even up to 91 % when testing swabs. This kind of clinical validation in field settings is a crucial step towards the application of point-of-care testing platforms. The successful application of the relatively new RPA technology is an exciting finding. However, the scientific rigor could be increased, by including additional controls and sample size calculations. Further, statistical analyses should be included to support the findings and conclusions (major revision). Overall, advancing molecular diagnostic methods towards field applicability is not only of interest to researchers in the field but due to the potential to translate to other pathogens also of value for other researchers.

PLOS authors have the option to publish the peer review history of their article (what does this mean?). If published, this will include your full peer review and any attached files.

Do you want your identity to be public for this peer review? For information about this choice, including consent withdrawal, please see our Privacy Policy.

Reviewer #1: No

Reviewer #2: No

Reviewer #3: No

Figure resubmission:

While revising your submission, we strongly recommend that you use PLOS’s NAAS tool (https://ngplosjournals.pagemajik.ai/artanalysis) to test your figure files. NAAS can convert your figure files to the TIFF file type and meet basic requirements (such as print size, resolution), or provide you with a report on issues that do not meet our requirements and that NAAS cannot fix. After uploading your figures to PLOS’s NAAS tool - https://ngplosjournals.pagemajik.ai/artanalysis, NAAS will process the files provided and display the results in the "Uploaded Files" section of the page as the processing is complete. If the uploaded figures meet our requirements (or NAAS is able to fix the files to meet our requirements), the figure will be marked as "fixed" above. If NAAS is unable to fix the files, a red "failed" label will appear above. When NAAS has confirmed that the figure files meet our requirements, please download the file via the download option, and include these NAAS processed figure files when submitting your revised manuscript.
---

## [Decision Letter · Decision Letter 1]

16 Mar 2026

PNTD-D-25-01865R1

Field deployment of a mobile suitcase laboratory for Buruli ulcer diagnosis in Ghana

Dear Dr. Frimpong,

Thank you for submitting your manuscript to PLOS Neglected Tropical Diseases. After careful consideration, we feel that it has merit but does not fully meet PLOS Neglected Tropical Diseases's publication criteria as it currently stands. Therefore, we invite you to submit a revised version of the manuscript that addresses the points raised during the review process.

We look forward to receiving your revised manuscript.

Kind regards,

Anil Fastenau, M.D., M.Sc.

Guest Editor

Mathieu Picardeau

Section Editor

Shaden Kamhawi

co-Editor-in-Chief

Paul Brindley

co-Editor-in-Chief

Journal Requirements:

1) We note that your Data Availability Statement is currently as follows: "All data are presented within the submitted manuscript.". Please confirm at this time whether or not your submission contains all raw data required to replicate the results of your study. Authors must share the “minimal data set” for their submission. PLOS defines the minimal data set to consist of the data required to replicate all study findings reported in the article, as well as related metadata and methods (https://journals.plos.org/plosone/s/data-availability#loc-minimal-data-set-definition).

If your submission does not contain these data, please either upload them as Supporting Information files or deposit them to a stable, public repository and provide us with the relevant URLs, DOIs, or accession numbers. For a list of recommended repositories, please see https://journals.plos.org/plosone/s/recommended-repositories

2) Please amend your detailed Financial Disclosure statement. This is published with the article. It must therefore be completed in full sentences and contain the exact wording you wish to be published.

1) State the initials, alongside each funding source, of each author to receive each grant. For example: "This work was supported by the National Institutes of Health (####### to AM; ###### to CJ) and the National Science Foundation (###### to AM).".

Reviewers' comments:

Reviewer's Responses to Questions

Key Review Criteria Required for Acceptance?

Methods

-Are the objectives of the study clearly articulated with a clear testable hypothesis stated?

-Is the study design appropriate to address the stated objectives?

-Is the population clearly described and appropriate for the hypothesis being tested?

-Is the sample size sufficient to ensure adequate power to address the hypothesis being tested?

-Were correct statistical analysis used to support conclusions?

-Are there concerns about ethical or regulatory requirements being met?

Reviewer #3: (No Response)

Reviewer #4: The objectives of the study are clearly articulated, focusing on evaluating the feasibility and diagnostic performance of a mobile suitcase laboratory platform using a recombinase polymerase amplification (RPA) assay for the field diagnosis of Buruli ulcer. The study design is appropriate for the stated objectives, as it evaluates the performance of the diagnostic platform directly in endemic clinical settings and compares results with the reference IS2404 qPCR assay performed at a reference laboratory.

The study population, consisting of clinically suspected Buruli ulcer patients recruited from three endemic districts in Ghana, is clearly described and relevant to the diagnostic question being addressed. The inclusion of both ulcerative and non-ulcerative lesions and the use of both swab and fine-needle aspirate samples provide a realistic representation of clinical practice in endemic settings.

While the sample size is relatively modest (73 participants), it appears adequate for a feasibility and field validation study and the authors provide justification for the sample size estimation. Statistical analyses including sensitivity, specificity, and predictive value calculations using contingency tables are appropriate for the evaluation of diagnostic performance.

Ethical considerations appear to have been adequately addressed. The manuscript reports approval from the Committee on Human Research, Publication and Ethics at the Kwame Nkrumah University of Science and Technology, and informed consent procedures are clearly described for both adults and children. Overall, the methodology is appropriate and sufficiently described to support the objectives of the study.

Reviewer #5: -Are the objectives of the study clearly articulated with a clear testable hypothesis stated? Yes

-Is the study design appropriate to address the stated objectives? Yes

-Is the population clearly described and appropriate for the hypothesis being tested? No

-Is the sample size sufficient to ensure adequate power to address the hypothesis being tested? No

-Were correct statistical analysis used to support conclusions? Yes

-Are there concerns about ethical or regulatory requirements being met? No

Results

-Does the analysis presented match the analysis plan?

-Are the results clearly and completely presented?

-Are the figures (Tables, Images) of sufficient quality for clarity?

Reviewer #3: - Fig 4: the background of this figure is too dark, the axis description for the y-axis is missing (assuming it is minutes?)

Reviewer #4: The analyses presented are consistent with the study objectives and described analytical approach. Diagnostic performance metrics including sensitivity, specificity, positive predictive value, and negative predictive value are clearly presented and appropriately stratified by specimen type.

The results demonstrate that the mobile BU-RPA platform achieved an overall sensitivity of 82% and specificity of 100% when compared with IS2404 qPCR, which provides a reasonable level of diagnostic performance under field conditions. The stratified analysis highlighting higher sensitivity in swab samples compared with FNA samples is informative and clinically relevant.

The turnaround time analysis clearly demonstrates the operational advantage of the platform, showing that diagnosis can be completed within approximately 45 minutes in field settings, compared with significantly longer turnaround times for centralized PCR testing.

Figures and tables appear adequate to illustrate the workflow and diagnostic performance of the system. The workflow diagram and field deployment images effectively demonstrate the practical implementation of the mobile laboratory platform.

Reviewer #5: -Does the analysis presented match the analysis plan? Yes

-Are the results clearly and completely presented? No

-Are the figures (Tables, Images) of sufficient quality for clarity? Yes

Conclusions

-Are the conclusions supported by the data presented?

-Are the limitations of analysis clearly described?

-Do the authors discuss how these data can be helpful to advance our understanding of the topic under study?

-Is public health relevance addressed?

Reviewer #3: Thank you to the authors for considering the feedback and adapting the manuscript accordingly. There are still some minor changes needed (e.g. adapting readability of figure 4), but the quality of the manuscript has increased substantially.

Reviewer #4: The conclusions are generally supported by the data presented. The authors appropriately conclude that the mobile suitcase laboratory platform enables rapid molecular confirmation of Buruli ulcer in endemic settings and has the potential to facilitate decentralized diagnostic capacity.

The manuscript appropriately discusses the main limitations of the study, including the moderate sensitivity observed in FNA samples and the relatively small sample size. The discussion also provides reasonable explanations for reduced sensitivity in certain specimen types and identifies potential operational considerations.

The study has clear public health relevance, particularly for improving access to timely diagnosis in Buruli ulcer endemic areas where laboratory infrastructure is limited. The concept of mobile molecular diagnostic platforms aligns with broader global health priorities for strengthening diagnostic capacity for neglected tropical diseases.

Reviewer #5: -Are the conclusions supported by the data presented? Yes

-Are the limitations of analysis clearly described? No

-Do the authors discuss how these data can be helpful to advance our understanding of the topic under study? Yes

-Is public health relevance addressed? Yes

Editorial and Data Presentation Modifications?

Reviewer #3: minor remarks:

- line 143: seventy-three instead of seventy- three

- line 183: sentences do not fit together

- line 273: remove b at end of line

Reviewer #4: The manuscript is generally well structured and clearly written. A few minor clarifications could further improve clarity:

• The discussion could briefly clarify the operational advantages of the RPA-based platform relative to previously described portable molecular diagnostic systems, particularly portable qPCR platforms referenced in the manuscript.

• The authors may consider briefly expanding the discussion on potential strategies to improve sensitivity for FNA samples.

• A short statement highlighting the level of training required for personnel to operate the mobile laboratory platform would further strengthen the implementation perspective.

These are relatively minor clarifications and do not affect the overall conclusions of the study.

Reviewer #5: (No Response)

Summary and General Comments

Reviewer #3: - IS2404 is a multi-copy target, could you indicate to how many bacteria the detection limit of 45 copies corresponds to?

- it is still not clear if the samples for field diagnosis and qPCR were taken from the same lesion, please indicate in the methods

-line 343: How do the authors expect the costs to reduce by a 10-fold?

- the authors could indicate more clearly in the conclusions that this promising setup still needs some tweaking before large-scale implementation

Reviewer #4: This manuscript presents a practical and well-executed evaluation of a mobile suitcase laboratory platform for the field diagnosis of Buruli ulcer using recombinase polymerase amplification. The study addresses an important operational challenge in neglected tropical disease control, namely the limited availability of molecular diagnostic capacity in endemic regions.

The work is timely and relevant to the goals of improving decentralized diagnostic capacity for skin neglected tropical diseases. The field-based validation conducted across multiple endemic districts strengthens the relevance of the findings. The results demonstrate that the mobile platform can deliver rapid diagnostic results within approximately 45 minutes while maintaining good specificity and acceptable sensitivity under field conditions.

Although the sample size is modest, the study provides valuable proof-of-concept evidence supporting the feasibility of deploying portable molecular diagnostic systems in endemic clinical settings. The manuscript is generally well written, the methodology is sound, and the conclusions are supported by the data presented.

Overall, the study represents a meaningful contribution to the field of point-of-care diagnostics for neglected tropical diseases. With minor clarifications as suggested above, the manuscript would be suitable for publication.

Reviewer #5: This manuscript evaluates a recombinase polymerase amplification (RPA) assay integrated into a mobile suitcase laboratory platform for the diagnosis of Buruli Ulcer in clinical settings. The manuscript is generally structured and the topic is relevant for improving early detection of the disease. However, several methodological and reporting issues require clarification before the conclusions regarding diagnostic performance and field applicability can be fully supported.

Overall Comment

The study mentions field deployment; however, the assay was conducted at clinics by skilled molecular biologists rather than minimally trained healthcare workers. The feasibility of healthcare workers (HCWs) performing the assay was not evaluated, and a cost–benefit analysis is needed before claiming field readiness. It would be more appropriate to describe the mobile suitcase platform and emphasize its potential for future field application rather than claiming immediate field readiness.

Specific Comments

Abstract

1.The abstract reports overall sensitivity and specificity but does not indicate that diagnostic performance differed between sample types (swab vs. FNA). Consider briefly mentioning this difference to provide a more balanced summary.

Introduction

2.Please clarify the specific advantages of RPA over other field-friendly molecular methods such as LAMP or portable PCR platforms.

Methods

3.The study design should be explicitly stated in the Methods section.

4.Please describe the DNA extraction method and the PCR procedure used as the gold standard.

5.Please clarify whether PCR and RPA were performed on the same specimen or on separate samples collected from the same lesion.

6.Since PCR and RPA results are available for the same participants, the authors may consider reporting an agreement analysis (e.g., McNemar test or kappa statistic).

7.Please clarify how the turnaround time was determined.

8.Please clarify how the assay cost was calculated and what components were included.

Results

9.Patient demographic characteristics should be clearly presented.

10.The sample size calculation required 49 PCR-positive cases, but only 39 PCR-positive cases were included. This deviation should be acknowledged as a limitation.

11.If positivity rates differed between PCR and RPA, this should be explicitly reported.

12.The subgroup sensitivity analyses (particularly for FNA samples) are based on relatively small sample sizes, resulting in wide confidence intervals. The authors may consider reporting positivity rates instead.

Discussion

13.The reasons why the RPA assay demonstrated lower sensitivity compared with the previous study and the Biomeme system should be discussed.

14.The reported $10 per test cost requires clarification regarding the cost components included in this estimate.

Tables / Presentation

14.A head-to-head comparison table of PCR, Biomeme, and the RPA assay (e.g., test cost, time to result, setup cost, infrastructure requirements, and operating conditions) would help clarify the comparative advantages of the proposed platform.

Language

16.The manuscript contains several typographical and grammatical errors (e.g., “laboratoy,” “polymerse,” “diganosising”). Careful proofreading is recommended.

PLOS authors have the option to publish the peer review history of their article (what does this mean?). If published, this will include your full peer review and any attached files.

Do you want your identity to be public for this peer review? For information about this choice, including consent withdrawal, please see our Privacy Policy.

Reviewer #3: No

Reviewer #4:  Yes: Sundeep Chaitanya Vedithi

Reviewer #5: No

Figure resubmission:
---

## [Editor Report · Decision Letter 2]

20 Apr 2026

Dear Dr Frimpong,

We are pleased to inform you that your manuscript 'Field deployment of a mobile suitcase laboratory for Buruli ulcer diagnosis in Ghana' has been provisionally accepted for publication in PLOS Neglected Tropical Diseases.

Best regards,

Anil Fastenau, M.D., M.Sc.

Guest Editor

Mathieu Picardeau

Section Editor

Shaden Kamhawi

co-Editor-in-Chief

Paul Brindley

co-Editor-in-Chief

---

## [Editor Report · Acceptance letter]

Dear Dr Frimpong,

We are delighted to inform you that your manuscript, "Field deployment of a mobile suitcase laboratory for Buruli ulcer diagnosis in Ghana," has been formally accepted for publication in PLOS Neglected Tropical Diseases.

Best regards,

Shaden Kamhawi

co-Editor-in-Chief

Paul Brindley

co-Editor-in-Chief
